# Evolving Applications of Circulating Tumor DNA in Merkel Cell Carcinoma

**DOI:** 10.3390/cancers15030609

**Published:** 2023-01-18

**Authors:** Varsha Prakash, Ling Gao, Soo J. Park

**Affiliations:** 1Division of Hematology and Oncology, University of California, San Diego, CA 92037, USA; 2Department of Dermatology, University of California, Irvine, CA 92697, USA

**Keywords:** Merkel cell carcinoma, ctDNA, liquid biopsy

## Abstract

**Simple Summary:**

Merkel cell carcinoma (MCC) is an aggressive neuroendocrine cutaneous malignancy that is well-suited for liquid biopsy due to high levels of tumor DNA shedding into the bloodstream. An increasing number of studies have illustrated potential applications of circulating tumor DNA (ctDNA) in the management of patients with MCC. Here we describe the evolving applications of ctDNA in MCC, from quantifying tumor burden, to monitoring for early recurrence, detecting minimal residual disease, and predicting treatment response.

**Abstract:**

Circulating tumor DNA (ctDNA) is a subset of circulating cell-free DNA released by lysed tumor cells that can be characterized by its shorter strand length and tumor genome-specific information. The relatively short half-life of ctDNA allows it to provide a real-time measure of tumor burden which has potential prognostic and surveillance value as a tumor biomarker. Merkel cell carcinoma (MCC) is a rare neuroendocrine skin cancer that requires close monitoring due to the high risk of relapse. There are currently no good tumor biomarkers for MCC patients, especially those who are negative for Merkel cell polyomavirus. ctDNA shows promise for improving the prognoses of MCC patients by monitoring tumor burden, identifying minimal residual disease (MRD), and stratifying patients by their likelihood of response to immune checkpoint inhibition or risk of relapse. In particular, bespoke ultra-sequencing platforms allow for the creation of patient-specific mutation panels that improve ctDNA detection, especially for patients with rare or uncharacteristic mutations. Leveraging bespoke ctDNA assays may improve physicians’ ability to alter treatment plans for non-responsive or high-risk patients. In addition, ctDNA MRD monitoring may allow physicians to treat relapses early before clinically evident disease is present.

## 1. Overview of ctDNA

Circulating cell-free DNA (cfDNA) was first described in 1948 by Mandel and Metais, who found nucleic acid and fragments of DNA that are present outside of cells and can be detected within body fluid [1]. In blood plasma, cfDNA consists of double-stranded DNA fragments of around 140–170 base pairs (bp), and mostly originates from leukocytes [2]. Circulating tumor DNA (ctDNA) is a subset of cfDNA released by lysed tumor cells that can be characterized by its shorter strand length and tumor genome-specific information [3]. The relatively short half-life of ctDNA allows it to provide a real-time measure of tumor burden which has potential prognostic and surveillance value as a tumor biomarker [4,5].

Liquid biopsy can be used to analyze circulating tumor cells (CTCs), cfDNA, and ctDNA through a simple blood test [6]. This method is minimally invasive and much more convenient for patients than frequent imaging for tumor surveillance [5]. The mutational profile from liquid biopsy also shows agreement between alterations in tumor and ctDNA [7]. Moreover, because ctDNA is genomic tumor DNA, this method overcomes issues of sampling tumor heterogeneity [8].

Analysis of ctDNA sampled through liquid biopsy may be tumor-agnostic or tumor-informed. The tumor-agnostic method only analyzes plasma samples and does not require a priori sequencing [5]. Examples of this testing method include Guardant360, FoundationOne Liquid, and Tempus xF, which all use hotspot mutation panels to identify tumor alterations in liquid biopsy [9,10]. The size of these panels can range from 55 to 311 genes, and they provide relatively quick and efficient ways of assessing ctDNA [9,10]. In contrast, tumor-informed analysis uses whole-exome sequencing to create bespoke mutation panels for each patient [5,7]. The Signatera platform involves whole-exome sequencing of tumor and matched normal blood samples to identify patient-specific alterations [11]. A bespoke panel of 16 single nucleotide variants (SNVs) is then selected to perform ultra-deep sequencing of patients’ blood for detection of ctDNA during surveillance [11]. This individualized approach is more time intensive than the tumor-agnostic method, but it does offer increased sensitivity, which is important for patients with rare or uncharacteristic mutations [5,7].

## 2. Applications of ctDNA

Circulating tumor DNA is a potential liquid-based biomarker for post-surgical surveillance and early detection of minimal residual disease (MRD). In the context of colorectal cancer (CRC), ctDNA has been studied as a predictive biomarker of response to adjuvant chemotherapy [5]. Henriksen et al. assessed the utility of using serial ctDNA sampling to predict response to adjuvant chemotherapy and relapse in 160 Stage III CRC patients [12]. Twenty patients were ctDNA-positive after surgery, and ctDNA samples before and after adjuvant chemotherapy were collected for 13 of these patients [12]. Of these 13 patients, only 3 showed clearance of ctDNA after receiving adjuvant chemotherapy and did not have disease recurrence within 36 months [12]. In contrast, all 10 remaining patients with no or transient ctDNA clearance relapsed within 36 months of follow-up after receiving adjuvant chemotherapy [12]. This demonstrates the potential value of using ctDNA to monitor treatment response in post-operative settings. In addition, the prospective GALAXY study monitored MRD to understand the relationship between ctDNA levels and efficacy of adjuvant chemotherapy in Stage I-III and oligometastatic Stage IV CRC patients [5]. Of 1000 patients with CRC, 188 were MRD-positive. Ninety-five of these patients received adjuvant chemotherapy and were shown to have a ctDNA clearance rate of 68% [5]. Those who did not receive adjuvant chemotherapy had a ctDNA clearance rate of only 7%, demonstrating that ctDNA can be used to assess therapy response in patients with MRD [5].

## 3. Merkel Cell Carcinoma and Current Biomarkers for Monitoring Disease

MCC is a rare neuroendocrine skin cancer that requires close monitoring due to the high risk of relapse [13]. This aggressive cancer often presents with nodal involvement or metastatic disease at the time of diagnosis, and mortality rates range from 33–46% [13]. MCC also has variable programmed death-ligand 1 (PD-L1) expression and tumor mutational burden (TMB), which limits treatment options for both primary and recurrent disease [14].

In the United States, it is estimated that up to 80% of MCC cases are positive for the Merkel Cell Polyomavirus (MCPyV) as measured by immunohistochemical identification. In these virus-positive cases, MCPyV oncoprotein antibody (AMERK) titers can be used to monitor disease progression, recurrence risk, and response to therapy [13,14]. However, it is crucial to establish baseline antibody titers within three months of surgery, since titers are expected to decrease significantly after clinically evident disease has been eliminated [13]. The remaining 20% of MCPyV-negative cases are primarily caused by UV damage and have a high TMB [13]. Intriguingly, a recent genomic study of 317 patients with MCC, the largest to date, was only able to detect the MCPyV genome in 114 cases (36%) [15]. This suggests a need for widespread testing of somatic DNA alterations rather than relying on surrogate testing assays such as AMERK to establish viral positivity, especially since MCPyV can be detected in normal skin and viral antibody can be detected in 55–87% of healthy individuals [16].

Although AMERK tests may be used to monitor MCC patients with MCPyV, there are few biomarkers currently available for MCPyV-negative cases. Both PD-L1 and TMB have been shown to be unreliable predictors of clinical outcomes [6,13,17]. Other potential biomarkers, such as serum neuron-specific enolase and synaptophysin, have been shown to have contradictory prognostic value for MCC patients [18]. Gambichler et al. characterized a subset of MCPyV-negative MCC tumors by low-level expression of mismatch repair proteins, which may be used as a future biomarker to stratify patients by potential response to immunotherapy [19]. In another study, this author also found that the pan-immune inflammation value of MCC patients is positively correlated with disease stage and may be used to predict disease recurrence [20]. While MCPyV-negative cases harbor a high percentage of somatic mutations in tumor suppressor genes such as *TP53* and RB1, these mutations span a wide spectrum and may occur outside hotspot regions which are not covered by tumor agnostic testing. Such mutations would be missed, thus supporting the need for personalization and ultra-deep sequencing of bespoke platforms.

## 4. Merkel Cell Carcinoma and ctDNA

Circulating tumor DNA has the potential to improve MCC patient outcomes by serving as a robust biomarker for disease recurrence, therapy response, and MRD. One major factor contributing to the high mortality rate in MCC is a five-year recurrence rate of roughly 40% [6]. Detecting recurrence earlier can improve prognoses, but no reliable biomarkers currently exist to identify early disease [6]. Moreover, there are no approved therapeutic alternatives for patients who are ineligible for or resistant to immune checkpoint inhibitors (ICI) [7]. Biomarkers to assess ICI response and monitor MRD would significantly improve patient outcomes by allowing clinicians to tailor therapies at different timepoints in the cancer trajectory. The scope of ctDNA detection has been significantly increased by modalities such as the Signatera MRD test, which employs targeted sequencing of personalized SNVs at ultra-deep levels (median target coverage, ≥105,000×). Thus, ctDNA may serve as a biomarker to predict treatment response to ICIs and beyond.

Shalhout et al. showed that cfDNA can be used as a diagnostic tool that detects disease and tracks early relapse or progression in MCC [6]. This study focused on 16 MCC patients who underwent cfDNA sequencing with an expanded cancer-associated gene panel. It was found that detection of cfDNA correlated with active MCC at the time of sample acquisition, and liquid biopsy was able to detect cfDNA in a patient before disease was evident on cross-sectional imaging [6]. This highlights the utility of cfDNA to predict early asymptomatic relapse in MCC patients who have received definitive therapy. However, the accuracy of this method may be improved by focusing more on ctDNA than cfDNA. In addition, the use of bespoke sequencing panels such as those in Signatera MRD assays would add accuracy and confidence in the detection of tumor burden and activity.

Although Shalhout et al. examined cfDNA in MCC, the first application of bespoke ctDNA in MCC was to monitor treatment response in a patient whose disease had progressed on pembrolizumab, a programmed cell death-1 (PD-1) inhibitor [7]. In a case report by Yeakel et al., ctDNA was found to be correlated with tumor burden and response to treatment [7]. The report focused on a 70-year-old woman with MCC of the left wrist with lymph node involvement who experienced progressive disease and severe adverse effects while on pembrolizumab [7]. The patient was found to have in-transit metastases of the left arm despite additional surgery and radiation [7]. She was subsequently treated with talimogene laherparepvec (T-VEC) injections and hypofractionated radiation therapy (HRT) [7]. Periodic bespoke ctDNA assays were used to monitor the patient’s disease burden and response to T-VEC and HRT [7]. Whole exome sequencing was then used to identify tumor-specific SNVs, and ultra-deep sequencing was performed to track SNVs in plasma ctDNA [7]. Serial ctDNA assays were shown to track with the patient’s disease course [7]. An initial positive liquid biopsy and 170-fold increase in ctDNA levels correlated with early metastases and the presence of new lesions on the patient’s arm [7]. After HRT was completed, ctDNA levels dropped significantly and eventually became undetectable after seven T-VEC injections [7]. As of follow-up day 441, the patient was off treatment without any signs of disease recurrence and had negative serial ctDNA assays [7]. Although this study was of a single patient, it highlights the value of serial ctDNA to not only monitor tumor burden but also follow treatment response in real time. This may be very beneficial in cases where recurrence is suspected but not established, and first-line therapy has shown to be ineffective. The use of ctDNA to gauge treatment selection beyond immunotherapy is especially important because second-line therapies for advanced MCC beyond ICI are limited.

Bespoke ctDNA has also been found to be sensitive in detecting MRD after treatment with curative intent as well as early relapse, which permits early therapeutic intervention [17]. In a pilot study of 30 MCC patients, Park et al. assessed 195 whole blood samples from 30 MCC patients with a maximum follow-up of 19 months [17]. Whole exome sequencing of tumor and matched normal blood was performed to identify tumor-specific SNVs, allowing for ultra-deep sequencing of plasma ctDNA [17]. These authors identified three ctDNA-positive patients, all of whom relapsed after definitive treatment [17]. Moreover, high ctDNA levels were found to be correlated with large tumor burden and metastatic disease [17]. Of note, elevated ctDNA in two patients led to initiation of early ICI therapy with rapid treatment responses [17]. Disease recurrence was identified in these patients prior to scheduled imaging studies, demonstrating how surveillance with ctDNA can improve clinical outcomes for MCC patients [17]. Fifteen patients had serial negative ctDNA tests with no evidence of disease recurrence after receiving definitive therapy [17]. Ten patients had initially undetectable ctDNA levels, and 7 of these patients had excellent responses to treatment [17]. Three patients did not respond favorably to ICI and displayed increasing ctDNA levels despite initially being negative on liquid biopsy [17]. Thus, bespoke serial ctDNA testing may also be used to risk-stratify patients who are more or less likely to respond to ICIs. This would allow physicians to consider alternate therapy before disease is clinically evident.

In addition to being a marker of active disease, Akaike et al. showed that bespoke ctDNA can be used to monitor disease burden and predict risk of recurrence in patients with MCC [14]. In a prospective study analyzing 328 blood samples from 125 patients, this group assessed whether ctDNA can identify disease burden and recurrence using tumor-informed sequencing [14]. Whole exome sequencing was conducted on tumor samples and matched with normal blood to identify an SNV set for each patient [14]. This SNV set was then tracked in serial blood samples using a PCR-NGS ctDNA assay [14]. These authors found 47 patients (38%) with clinically significant MCC who were all ctDNA-positive at first sample acquisition [14]. For 24 patients who were newly diagnosed with MCC, primary tumor diameter was also found to be correlated with ctDNA value [14]. Seventy-three of the original 125 patients (58%) were enrolled in a surveillance arm to understand the benefits of serial ctDNA testing [14]. Seven out of 145 total ctDNA assays were positive, and 5 of these patients developed clinically relevant disease within 60 days of their first positive ctDNA test [14]. The estimated risk of recurrence within 60 days of a positive ctDNA assay was found to be 57%, while the risk after a negative test was 0% within 60 days and 3% from 60–90 days [14]. This study’s findings support the idea that ctDNA can be used as a marker of early recurrence in MCC patients and identify disease before it is clinically evident. These outcomes are in concordance with the findings by Park et al., who were able to initiate early treatment for MCC patients based off bespoke ctDNA testing [14]. Taken together, these studies highlight the potential for ctDNA to significantly improve outcomes and effectively combat recurrence for MCC patients through effective monitoring of MRD.

## 5. Conclusions and Future Directions

Merkel cell carcinoma is a highly aggressive skin cancer with high risk of recurrence after first-line therapy. Although most patients undergo imaging studies to monitor for disease recurrence, more sophisticated and convenient surveillance methods are needed to identify early recurrence before it manifests as clinically relevant disease. This is especially true for MCPyV-negative cases. In addition, predictive biomarkers are needed to monitor response to therapy and identify patients who are less likely to respond favorably to first-line ICI. The ease of testing and minimally invasive nature of liquid biopsy over other sampling methods facilitates more frequent analysis. Moreover, bespoke tumor-informed analysis of ctDNA allows for greater accuracy of results. This is crucial for variably mutated tumors such as MCC.

Circulating tumor DNA has the potential to be a valuable tool in the adjuvant setting. First, it can be used to identify high-risk patients who may benefit from adjuvant therapy. Second, serial ctDNA testing can be used to monitor response to therapy for patients with advanced disease. This is especially important for MCC patients, as there are limited alternatives to ICI and changes to treatment plans must be quickly implemented to avoid widespread disease dissemination. Serial ctDNA assays can also be used as a molecular tool to guide surveillance or therapy in MCC. For instance, serial ctDNA testing can be used to detect MRD in patients who have received treatment with curative intent. Circulating tumor DNA has also been shown to detect MRD earlier than conventional modalities such as physical exams and imaging, and thus may be used to initiate early therapy in patients before clinically relevant disease is identified. For patients with MCC, the future of ctDNA has now arrived.

## Data Availability

No new data were created or analyzed in this study. Data sharing is not applicable to this article.

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
