# Peer review of "Evolving Applications of Circulating Tumor DNA in Merkel Cell Carcinoma"

_cancers, 2023, doi:10.3390/cancers15030609_

Round 1

Reviewer 1 Report

This is a very well written, topical paper on the use of ctDNA for Merkel cell carcinoma.  The authors seem to indicate that the bespoke ctDNA platform is superior to a tumor agnostic approach.  However, for non-viral associated MCC, which rightly identified is in need of better biomarkers, the mutations are relatively conserved with p53 and rb mutations.  Why do we need a bespoke platform in this event as this would likely be captured in an off the shelf, tumor agnostic approach?  I would like to see this addressed or clarified in the manuscript.  Otherwise, very well done summarizing an emerging field. 

Reviewer 2 Report

The article deals with ctDNA in Merkel cell carcinoma, which could play a major role in tumour therapy in the future. CtDNA will further personalise oncological therapy and is an important topic. Only small changes are necessary.Language corrections
Line 65
“Henriksen et al.”Line 83, 86, Line 202
Abbreviations should be used at this point. “MCC”Corrections to the content
Line 103
MCPyV-negative MCC is characterised by low expression of mismatch repair proteins. This should be discussed as a possible approach or biomarker.
Gambichler T, Abu Rached N, Tannapfel A, Becker JC, Vogt M, Skrygan M, Wieland U, Silling S, Susok L, Stücker M, Meyer T, Stockfleth E, Junker K, Käfferlein HU, Brüning T, Lang K. Expression of Mismatch Repair Proteins in Merkel Cell Carcinoma. Cancers (Basel). 2021 May 21;13(11):2524. doi: 10.3390/cancers13112524Line 103
The prognostic ability of the blood-based biomarker PIV in MCC patients should be discussed here.
Gambichler T, Said S, Abu Rached N, Scheel CH, Susok L, Stranzenbach R, Becker JC. Pan-immune-inflammation value independently predicts disease recurrence in patients with Merkel cell carcinoma. J Cancer Res Clin Oncol. 2022 Nov;148(11):3183-3189. doi: 10.1007/s00432-022-03929-y
